# Partitioning of Metal Contaminants between Bulk and Fine-Grained Fraction in Freshwater Sediments: A Critical Appraisal

Neda Vdović *, Mavro Lučić , Nevenka Mikac and Niko Bačić

Division for Marine and Environmental Research, Ruđer Bošković Institute, Bijenička Cesta 54, 10000 Zagreb, Croatia; mlucic@irb.hr (M.L.); mikac@irb.hr (N.M.); nbacic@irb.hr (N.B.)
* Correspondence: vdovic@irb.hr; Tel.: +385-1-4561-176

**Abstract:** The distribution of six common metal contaminants (Cd, Cr, Cu, Ni, Pb, Zn) in the bulk (<2 mm) and fine fractions (<63 μm) of freshwater sediments was compared to conclude on the long-existing dilemma which fraction should be used in the investigation of the metal contamination. The environments included in the study (24 rivers, 8 lakes) were very different with respect to sediments origin and composition and they provided a good review of the possible scenarios. For the river sediments, particularly those having >40% of sand fraction, metal concentrations were up to seven times higher in the fine fraction, implying the necessity for considering sand dilution effect in compositional data analysis. The same samples were also characterized with higher organic matter content (OM) in the fine fraction. Lake environments were characterized by fine-grained sedimentation and the difference between metal concentrations in the bulk and fine fraction was not so expressed. The preparation of samples for the geochemical and compositional data mining should be carried out in accordance with the sedimentological characteristics of the investigated environment. It implies that the insight into geological setting and determination of sedimentological characteristics should be an obligatory part of monitoring/investigating metal contamination in freshwater sediments. For river sediments, the analysis of the fine sediment fraction or correction for sediment lithology are advisable.

**Keywords:** freshwater sediments; metal contaminants; bulk sediment; fine fraction; river



## 1. Introduction

Mineral particles in freshwater environments are transported downstream by the river flow depending on the particle size and intensity of the river discharge. In that process, coarser particles tend to be more stationary while fine material is easily resuspended and displaced. Therefore, the particle size distribution (PSD) of river sediments is quite variable. Likewise, the transport of contaminants associated with mineral particles is significantly affected by their particle size distribution [1].

Investigations focused on metal concentrations in different particle size fractions could provide more detailed information on their distribution mechanisms and ecological risk to aquatic environments. There are many studies on the relationship of heavy metal content and particle size distribution (PSD) in sediments [2–17] but conclusions obtained in these studies are very different. Some researchers indicated that metals tend to be present mainly in the fine sediment fraction [3–6,9] while others found no significant difference between metal content in the bulk and fine fraction [2,10]. The main argument was the formation of aggregates and metal-bearing organic matter and/or Fe/Mn-oxyhydroxide coatings on sand particles at locations affected by contamination [2] or the existence of the nearby mining areas which contribute to the accumulation of the certain element in the coarse fraction [11]. Likewise, in some studies metal concentrations were determined in fine fraction (<63 μm) [8,12], while the practice of separating the sand and performing

the geochemical analysis in the fine fraction was not employed in others [9,13]. Besides grain-size correction, the geochemical normalization is found to be a promising tool for the investigation of metal contamination in river sediments [14,15], particularly for studying a single river systems. However, for regional and national studies, encompassing very different geological settings, relevance of using common background and enrichment factors approaches (which include normalization) was questioned [16].

The attempt to find a suitable protocol for monitoring fluvial sediments, comparing bed and bank sediments and suspended material, was reported by Mokwe-Ozonzeadi et al. [17]. They found <2 mm sediment fraction to be the most suitable for the monitoring purposes of the gravel bed river mostly because it is difficult to collect sufficient mass of the <63 μm fraction without contaminating the sample. However, the procedure recommended by the Water Framework Directive [18], for the correction for grain size effects in river sediments, is the collection of the freshly deposited <63 μm sediment fraction or the suspended particulate material (SPM).

There is another important fact to point out—the studies listed above were mostly conducted in a single aquatic environment; studies comparing a large number of freshwater environments are not found in the literature. In an attempt to add some new insight in the matter, this study is aimed at: (1) Investigating the distribution of selected heavy metals (Cd, Cr, Cu, Ni, Pb, and Zn) in the bulk (<2 mm) and fine-grained fraction (<63 μm) of freshwater sediments from 24 rivers and 8 lakes; (2) evaluating the differences in metal concentration between those two fractions and their relation to sediment characteristics (particle size distribution and organic matter content); and (3) bringing the potential guidance for future environmental and monitoring studies assessing the metal contamination in freshwater sediments.

## 2. Materials and Methods

### 2.1. Sampling and Preparation of Samples

The sampling of the sediments was undertaken in the frame of the pilot monitoring program in Croatia. The purpose of the program was to establish the protocol for monitoring freshwater sediments and biota. Sampling was conducted in summer—early autumn 2019 during low river discharge. Samples were collected at 47 locations (Figure 1, Table S1), covering 32 freshwater environments (24 rivers, 6 lakes, and 2 artificial reservoirs). In larger rivers, samples were taken from more than one location. Geological setting of the whole area is shown in the Supplementary File (Figure S1).

The river sediments were taken with a spatula along the river channel at protected parts of the river flow where finer material could be found. At each location, at least three subsamples were taken and mixed to obtain the composite sample representative of the location. Lake sediments were sampled from a boat, using a Van Veen grab, in approximately central part of the lake. Immediately after sampling, sediments were washed through a 2 mm-mesh sieve using ambient river/lake water to separate gravel and large organic debris. The samples were then stored in plastic bags and transported to the laboratory where a small portion (2–5 g) of the wet sediment was separated for the purpose of particle size determination while the rest was air-dried and homogenized by means of a ball mill for other analyses.

Besides bulk sediment (<2 mm), fine fraction (<63 μm) was also collected in situ using 63 μm-mesh sieve. Sediments were sieved using ambient water to avoid chemical perturbation, and the suspensions containing <63 μm particles were stored in 1 L plastic bottles. In the laboratory, suspensions were left to settle, surplus water was carefully decanted, and the remaining material was lyophilized (Freezone 2.5, Labconco Corporation, Kansas City, MS, USA).

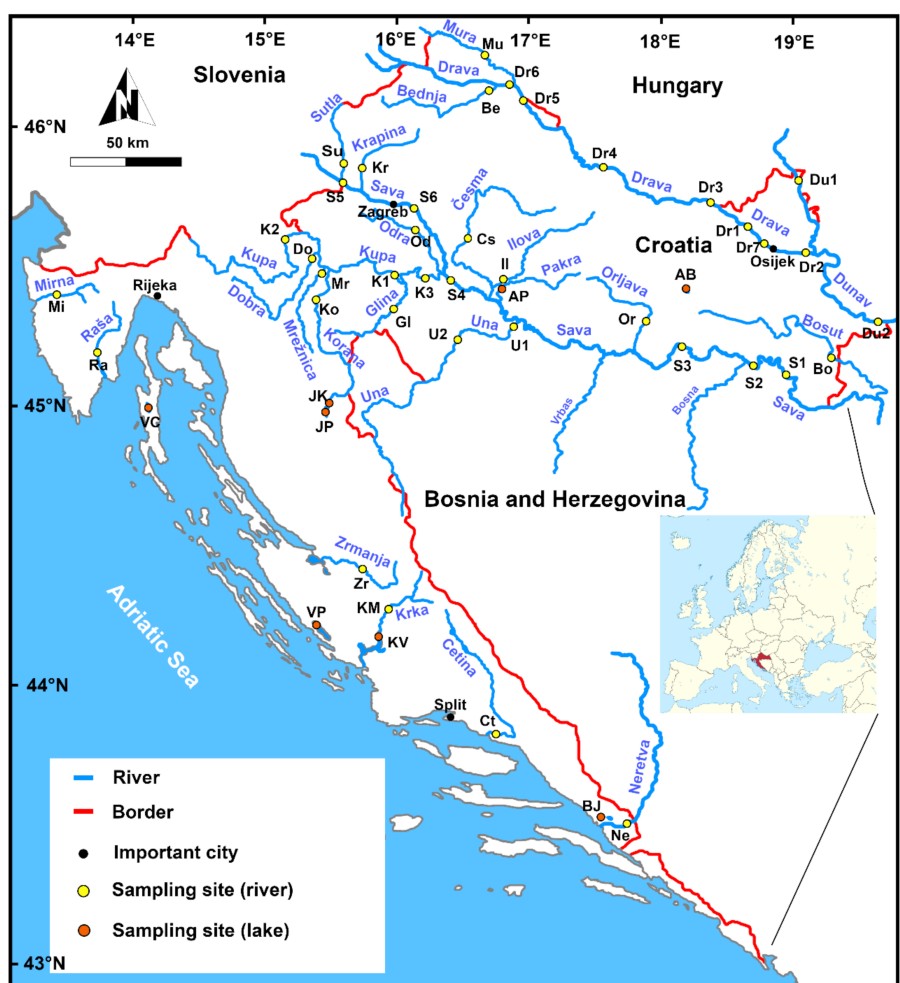

**Figure 1.** Sediment sampling locations.

### 2.2. Analyses

Particle size distribution (PSD) was determined using a laser-based particle size analyzer (LS 13320; Beckman Coulter Inc., Indianapolis, In, USA). The PSD was calculated using Mie theory of light scattering (optical parameters: refractive index = 1.53; absorption index = 0.1).

Organic matter (OM) content was estimated as a loss on ignition at 360 °C for 2 h [19].

Multi-elemental analysis was conducted using a high resolution inductively coupled plasma—mass spectrometer (HR ICP-MS), Element 2 (Thermo Finnigan, Bremen, Germany). Prior to analysis, samples were digested in a microwave oven (Multiwave 3000, Anton Paar, Graz, Austria) in a two-step procedure: I) 5 mL $HNO_3$ (65%, pro analysis, Kemika, Zagreb, Croatia) + 1 mL HCl (37%, VLSI Grade, Rotipuran, Carl Roth, Karlsruhe, Germany) + 1 mL HF (47–51%, for trace analysis, Fluka, Buchs, Switzerland); II) 6 mL $H_3BO_3$ (40 g $L^{-1}$, Fluka). After digestion samples were diluted ten-fold, acidified with 2% (v/v) $HNO_3$ (65%, s.p., Fluka) and indium (In, 1 µg $L^{-1}$) was added as the internal standard. Analytical quality control was performed by simultaneous analysis of procedural blanks and certified reference materials of soil NCS DC 773902 (GBW 7410) and stream sediment NCS DC 73309 (GBW 07311) for which good recoveries (90–100%) were obtained, depending on the element measured. Details of the method are provided elsewhere [20].

Principal component analysis (PCA) was performed to identify the main factors that govern the element distribution. To remove the non-negativity and constant-sum constraints on compositional variables, the centered log–ratio transformation (clr) [21] was previously applied. The clr transformation can be derived by dividing each compositional

variable (element) by the geometric mean of the composition and then taking the logarithm of each quotient. Following the clr transformation, PCA was performed using the free software environment [22].

## 3. Results and Discussion

### 3.1. The Particle Size Distribution (PSD)

The general overview of the PSD in the bulk sediment fraction, given in Figure 2 and Table S1, shows a wide variety of sample characteristics even for the samples pertaining to the same river (e.g., samples S1–S6 (Sava) and Dr1–Dr7 (Drava)). A significant number of samples was quite coarse-grained; nineteen of them had >50% of sand fraction. A rather high content of sand fraction in most samples is probably a consequence of strong hydrodynamics of these environments. Sampling during low discharge when a bed sediment remains undisturbed, allows fine material, usually composed of <63 μm-sized particles, to settle, but the sand still prevails in these high-energy environments. In addition, some of the fine-grained material may be flocculated [23] and attributed to the sand fraction since no chemical agent or physical force was applied during the wet sieving of the sediment samples.

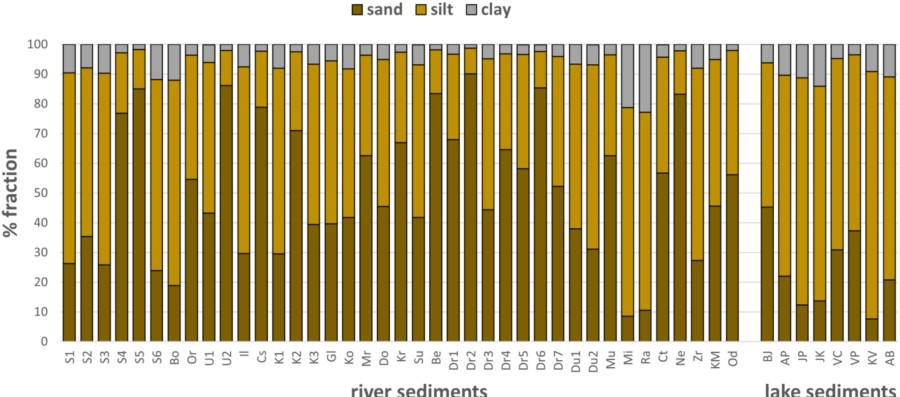

**Figure 2.** Particle size distribution of bulk sediments.

The lowest sand content was determined for samples Mi, Ra, JP, JK, and KV. Samples Mi and Ra pertain to the rivers Mirna and Raša which drain carbonate rocks and flysch deposits (Figure S1) of which marls and shales are prone to physico-chemical weathering while sandstones and conglomerates are more resistant [24]. Samples JP, JK, and KV are lake sediments deposited in low-energy environments of the Plitvice Lakes National park (JP and JK) [25,26] and the Visovac Lake (KV) in the Krka National park [27], and therefore, the lowest sand fraction was detected.

### 3.2. Organic Matter (OM)

The organic matter content in samples ranged between 0.51% and 10.56% in total sediment and between 2.59% and 9.20% in fine fraction (Table 1). The highest OM share (>10%) was determined in bulk samples 39 (Odra) and 45 (Lake Vransko-Biog). Somewhat lower, but still high concentrations were also determined in other lake sediments, particularly samples 41–44 (Resevoir-Pakra, Lake Prošće, Lake Kozjak, Lake Vransko-Cres), indicating reductive conditions in lake environments due to high primary production and stationary water [28,29]. Such conditions are not expected in river environments due to constant water flow. However, high OM content was determined in case of the Odra River. The area around the Odra serves as a floodplain for the excess water during high discharge of the Sava River [30]. Because of frequent flooding, the area assumes characteristics of a wetland—a damp area that enables the organic matter accumulation.

**Table 1.** The organic matter content in bulk and fine fraction of sediment samples.

| Sample No. | Abbreviation | Location | OM % Bulk Sed. | OM % Fine Fraction | Sample No. | Abbreviation | Location | OM % Bulk Sed. | OM % Fine Fraction |
|---|---|---|---|---|---|---|---|---|---|
| 1 | S1 | Sava-Županja | 5.43 | 5.08 | 25 | Dr3 | Drava-D Miholjac | 2.47 | 5.73 |
| 2 | S2 | Sava-Šamac | 4.16 | 5.18 | 26 | Dr4 | Drava-T Polje | 1.04 | 4.52 |
| 3 | S3 | Sava-Sl Brod | 4.16 | 4.13 | 27 | Dr5 | Drava-Botovo | 4.10 | 5.70 |
| 4 | S4 | Sava-Lukavec | 1.15 | 4.41 | 28 | Dr6 | Drava-Legrad | 0.54 | 4.47 |
| 5 | S5 | Sava-Drenje | 1.61 | 6.28 | 29 | Dr7 | Drava-Josipovac | 1.32 | 3.67 |
| 6 | S6 | Sava-Rugvica | 4.82 | 4.18 | 30 | Du1 | Dunav-Batina | 3.75 | 4.04 |
| 7 | Bo | Bosut | 2.45 | 2.59 | 31 | Du2 | Dunav-Ilok | 3.72 | 4.44 |
| 8 | Or | Orljava | 2.32 | 3.88 | 32 | Mu | Mura | 1.28 | 5.81 |
| 9 | U1 | Una-Jasenovac | 4.26 | 5.15 | 33 | Mi | Mirna | 3.62 | 3.27 |
| 10 | U2 | Una-Kostajnica | 1.98 | 6.40 | 34 | Ra | Raša | 3.98 | 2.91 |
| 11 | Il | Ilova | 3.28 | 3.74 | 35 | Ct | Cetina | 3.86 | 9.20 |
| 12 | Cs | Česma | 0.62 | 4.22 | 36 | Ne | Neretva | 1.90 | 6.13 |
| 13 | K1 | Kupa-Šišinec | 4.94 | 5.04 | 37 | Zr | Zrmanja | 2.50 | 3.28 |
| 14 | K2 | Kupa-Bubnjarci | 3.19 | 4.95 | 38 | KM | Krka-Manastir | 6.47 | 3.66 |
| 15 | K3 | Kupa-M Gorica | 1.31 | 5.33 | 39 | Od | Odra | 10.56 | 6.68 |
| 16 | Gl | Glina | 4.76 | 4.07 | 40 | BJ | Baćinska Lakes | 2.73 | 2.85 |
| 17 | Ko | Korana | 5.40 | 5.22 | 41 | AP | Reservoir-Pakra | 5.73 | 4.91 |
| 18 | Mr | Mrežnica | 6.14 | 6.24 | 42 | JP | Lake Prošće | 8.01 | 5.76 |
| 19 | Do | Dobra | 4.56 | 5.62 | 43 | JK | Lake Kozjak | 6.93 | 4.42 |
| 20 | Kr | Krapina | 1.60 | 6.37 | 44 | VC | Lake Vransko-Cres | 7.44 | 8.73 |
| 21 | Su | Sutla | 2.90 | 5.71 | 45 | VP | Lake Vransko-Biog | 10.26 | 8.21 |
| 22 | Be | Bednja | 0.83 | 5.47 | 46 | KV | Krka-Lake Visovac | 4.13 | 3.83 |
| 23 | Dr1 | Drava-Belišće | 1.53 | 5.82 | 47 | AB | Reservoir-Borovik | 3.85 | 3.17 |
| 24 | Dr2 | Drava-Dunav | 0.51 | 4.94 | | | | | |

Significant difference between the OM content in bulk sediment and fine fraction was observed in 15 river sediments (4, 5 (Sava), 10 (Una-Kostajnica), 12 (Česma), 15 (Kupa-M Gorica), 20 (Krapina), 21 (Sutla), 22 (Bednja), 23–29 (Drava), 32 (Mura), 36 (Neretva)). In all these samples, the OM in fine fraction was at least two times higher (the ratio ranged between 2 and 10) than in the bulk sediment. This result is not surprising since the organic matter prefers adsorbing to fine particles while common characteristic of these samples was high content of sand fraction.

There was also an opposite trend, lower OM in fine fraction, noticed in several samples (38 (Krka-Manastir), 39 (Odra), 41–43 (Reservoir-Pakra, Lake Prošće, Lake Kozjak). That change was not so strongly expressed (at the most by a factor of 1.7) and it was probably the consequence of sand fraction separation by which a particulate and/or flocculated organic matter might have been removed. The samples showing that trend were mostly lake environments or river systems with significant eutrophication [28,29].

### 3.3. Geochemical Composition

3.3.1. Major Elements

Aluminum is usually considered as an indicator of clay fraction in the sediment [14] and it is expected to be found in higher concentrations in fine-grained samples. It is commonly used for geochemical normalization, correction for the grain size effects on metal concentrations, and the assessment of the origin of metals in sediments [15]. To some extent there is a general agreement between the clay fraction and Al concentration in investigated sediments (Figure 3A), as well as between Al and fine fraction (Figure 3B), but some locations deviate significantly (K3 (Kupa-M Gorica), Ct (Cetina), Od (Odra), BJ (Baćinska Lakes), JP (Lake Prošće), JK (Lake Kozjak) and KV (Krka-Lake Visovac)). These samples pertain to lake environments with authigenic carbonate precipitation or rivers draining exclusively carbonate terrains [25,26,31], (Figure S1). Namely, authigenic carbonate precipitates as very fine particles [28,32] poor in Al. The exception makes sample K3 (Kupa-M Gorica) in which the lower content of aluminum (Figure 4A) and calcium (Figure 4B) and all contaminants (Figure 5) was determined. This sample was collected in the area characterized by quaternary alluvium and loess (Figure S1). Presumably, the presence of quartz component, accumulated in the coarse silt and sand, played an important role in diluting the chemical composition of bulk sediment fraction [33,34]. Samples Mi (Mirna) and Ra (Raša) deviate even more from the Al-clay regression line; the highest share of clay fraction determined in these samples does not coincide with the highest content of Al. As mentioned earlier, these samples belong to the rivers draining flysch deposits (Figure S1) where marls having high content of $CaCO_3$ contribute to the mineral composition of these sediments and dilute the Al-bearing clay component [35]. For all of the mentioned above, the normalization to Al did not seem suitable as a tool for the correction of the grain-size effects on metal concentration in the assessment of the possible anthropogenic impact on the investigated sediments. This conclusion is also supported by the lack of correlation between Al and investigated metal contaminants in the bulk and the fine sediment fraction which was tested for the given data set but not shown here.

Calcium concentrations in bulk sediments (Figure 4B), serving as a proxy for the $CaCO_3$ content in samples, corroborated aforesaid reasoning. The samples that stand out the most are the same ones with the lowest Al content (Ct, KM, BJ, JP, JK, and KV). However, these are not the only samples with significant Ca content—samples U1, U2 (Una), Ko, Mr, Do (Korana, Mrežnica, Dobra), Ct, Ne, Zr, Km, Od (Cetina, Neretva, Zrmanja, Krka, Odra) were taken from the rivers that drain karstic terrains; samples Mi and Ra, as earlier mentioned, originate from the Mirna and Raša, the rivers draining carbonate-rich flysch deposits (Figure S1), [35].

The highest Ca concentrations were determined in the group of lake sediments. Five of them (BJ, JP, JK, VC, VP, and KV) are the lakes incised in carbonate terrains in which authigenic $CaCO_3$ precipitation takes place [25–27,31]. Samples AP and AB are reservoirs situated in the northern part of Croatia characterized by clastic Neogene and Quaternary deposits of Pannonian basin (Figure S1), [36].

3.3.2. Metal Contaminants

Concentrations of potentially toxic elements (Cd, Cr, Cu, Ni, Pb, and Zn) in the bulk and fine fraction are shown in Figure 5. The results revealed quite different element concentrations in the investigated samples; the most notable was higher concentration of investigated elements in the fine fraction of river samples, particularly those having sand fraction share >40% (Figure 2). That difference was not observed in lake sediments since those environments are characterized by lower water energy and consequently higher content of fine-grained sediment in the bulk (Figure 2).

The sediment quality with respect to analyzed elements was assessed according to Norwegian sediment quality guidelines [37] in the absence of European or Croatian regulations. Bakke et al. [37] define five contamination classes: I—background level, II—no toxic effects, III—toxic effects after chronic exposure, IV—toxic effects after short term

exposure, V—severe acute toxic effects. The upper limit of class II denotes Predicted No Effect Concentrations (PNEC). The border between class II and class III is most significant as it separates no-effects sediments from those for which toxic effects could be expected.

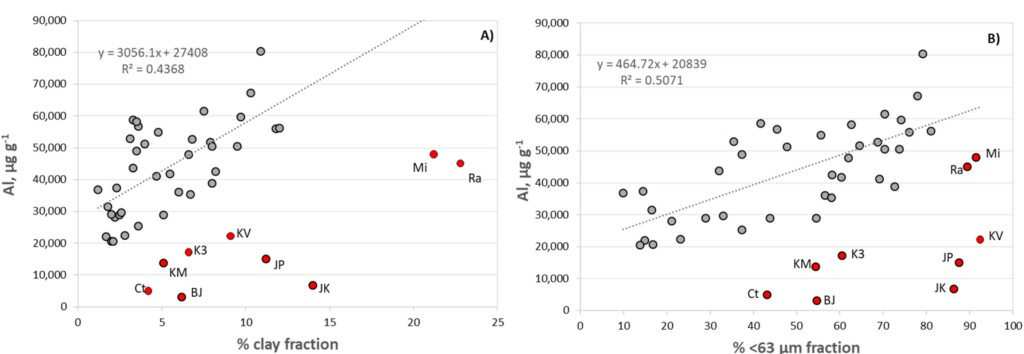

**Figure 3.** (**A**) Al concentration vs. clay fraction in sediment samples; (**B**) Al concentration vs. fine fraction in sediment samples. Samples marked red were omitted from the regression analysis. The *p*-value was set at *p* < 0.001 to indicate statistical significance (*p*-values were less than 0.000006).

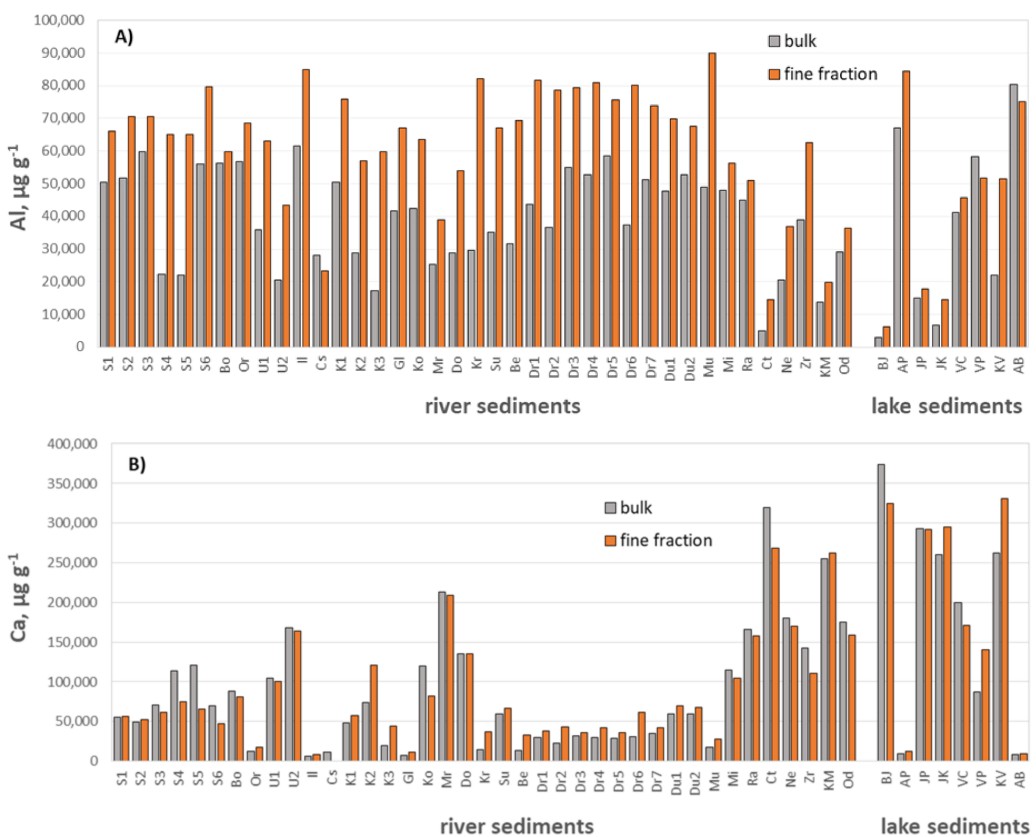

**Figure 4.** Distributions of Al (**A**) and Ca (**B**) in bulk and fine sediment fraction.

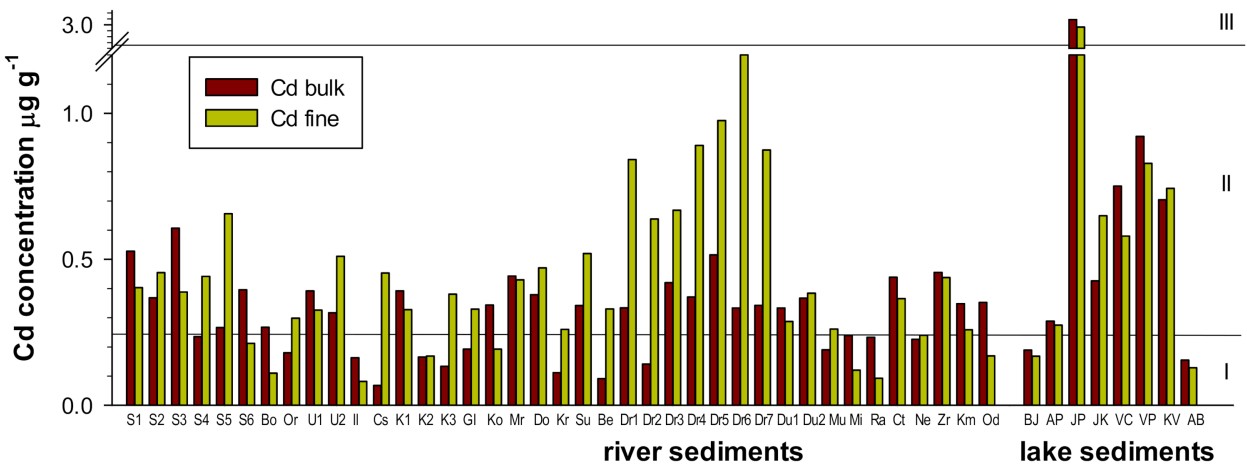

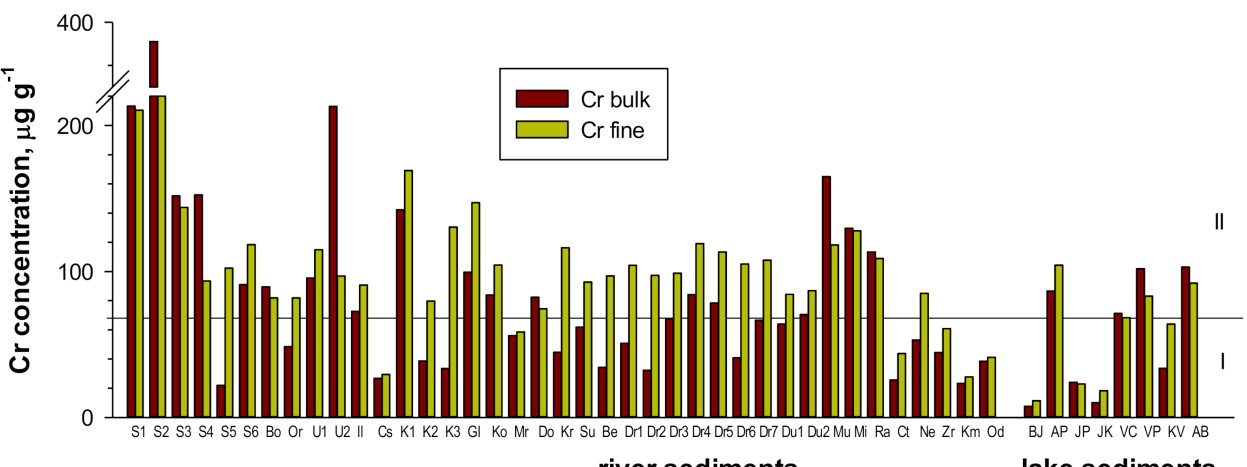

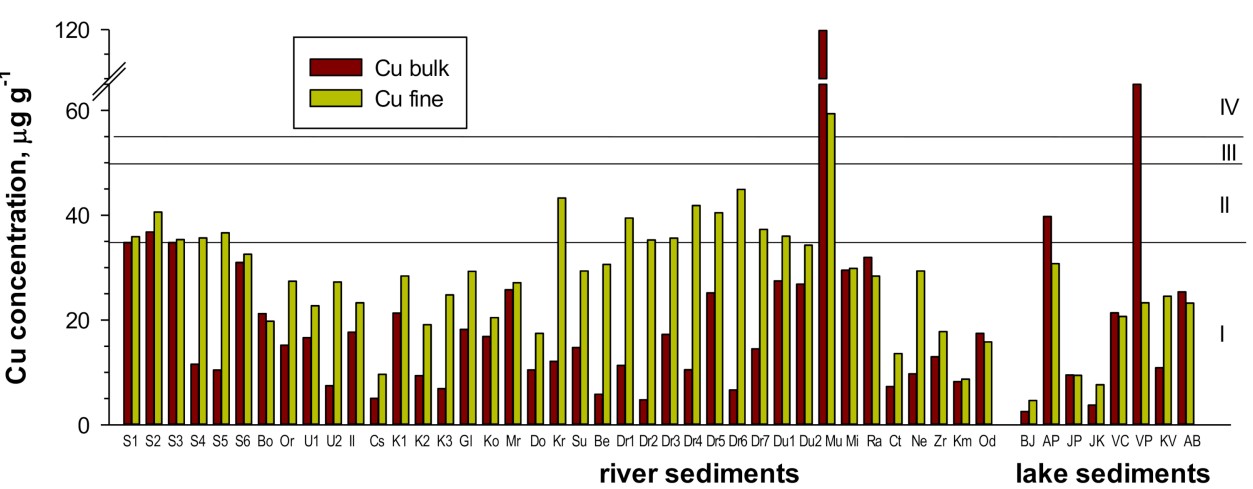

(**A**)

**Figure 5.** *Cont.*

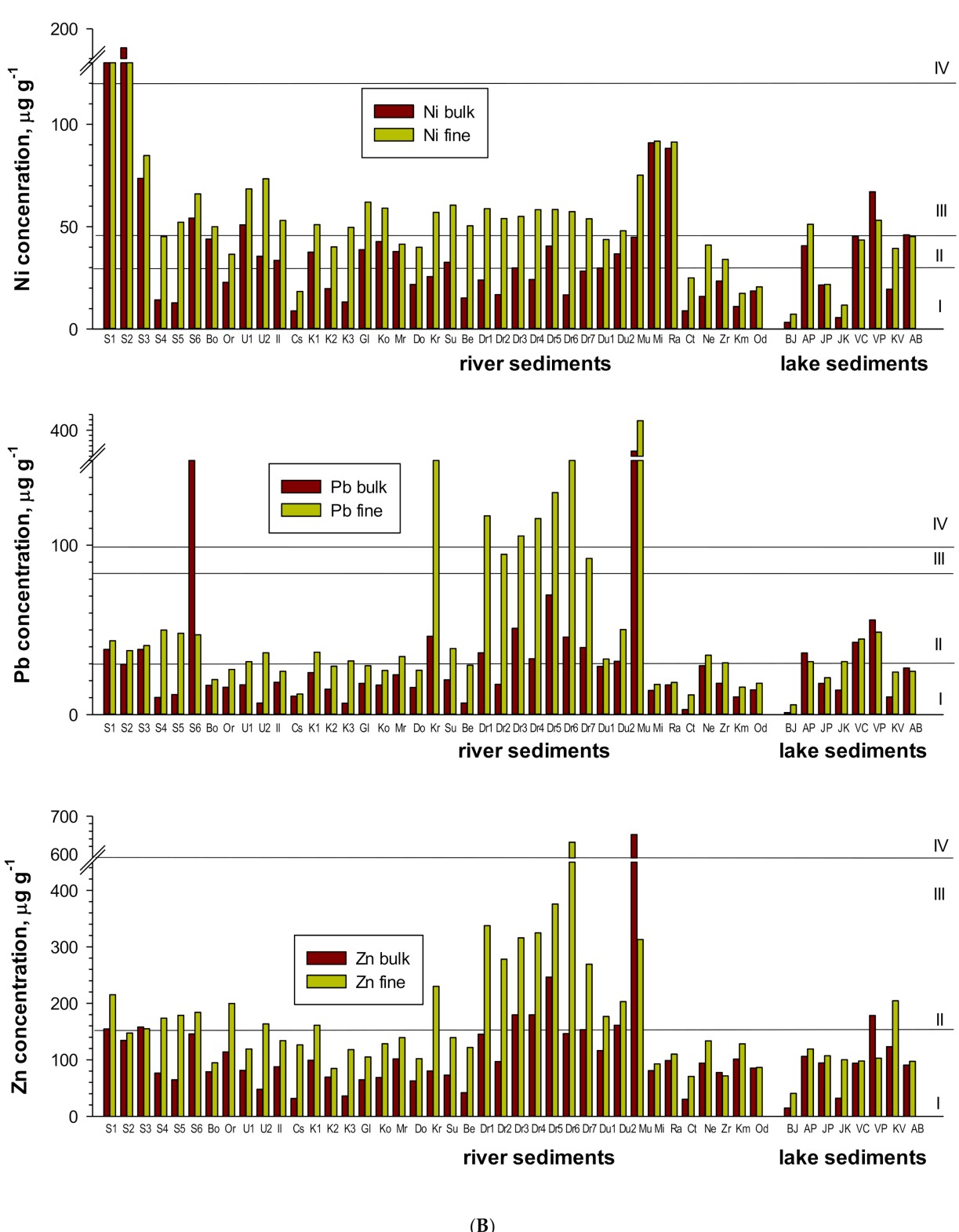

**Figure 5.** (**A**) Distributions of Cd, Cr, and Cu in bulk and fine fraction of sediments. Horizontal lines represent sediment quality guideline classes [37]. (**B**) Distribution of Ni, Pb, and Zn in bulk and fine fraction of sediments. Horizontal lines represent sediment guideline classes [37].

According to the mentioned quality criteria, the investigated samples are of Ist and IInd category with respect to Cd and Cr for both bulk and fine fraction. The exception is Cd in sample JP (Lake Prošće) which is of natural origin, related to Cd-rich dolostones [38].

The situation is rather similar for Cu and Zn—most of the samples could be sorted in Ist and IInd class for both analyzed sediment fractions. The exceptions are Cu concentrations in samples Mu (Mura) and VP (Lake Vransko-Biog) and Zn concentrations in samples D6 (Drava-Legrad) and Mu (Mura), all reaching class IV and indicating possible contamination with Cu and Zn. The highest concentrations of Pb (classes III and IV) were found in the fine fraction of samples pertaining to the River Drava, and in the bulk sample S6 (Sava-Rugvica). Approximately half of the samples reached class III with respect to Ni, particularly fine fraction, while samples S1 and S2 reached even class IV both in bulk and fine fraction. In all, the most affected of all investigated environments are the rivers Drava and Mura; mining and smelting activities conducted in that area since antiquity, significantly contributed to the chemical composition of the soil and sediment of the Drava Valley [39]. Anomalies observed in the Sava River are most probably related to the sediment supply from the right-side tributaries of the Sava River that drain the Central Dinaridic Ophiolitic Zone [40], abundant in ultramafic and metamorphic rocks. High levels of Cr and Ni in sediments of the Sava River at locations S1 and S2 were also observed in other studies and were ascribed to anthropogenic impact of the metal industry [41]. The same authors also showed that Cr and Ni exist primarily in the sparingly soluble forms.

Sediment quality guidelines used worldwide vary regarding the recommendation on which compartment (suspended matter or sediment) should be considered and which concentration thresholds are used to assess sediment quality [42]. Current proposal to the EC is that suspended sediments are more appropriate sampling mediums than the settled sediments for assessing the general risk of sediment-bound chemicals to wildlife [18,43]. Presented assessment of the quality of Croatian freshwater sediments showed some differences, depending on which fraction (fine or bulk) was used for evaluation. Results of our previous study [44] have shown that geochemical compositions of the SPM and fine sediment fraction of the Sava River were similar. By taking all that into account, we would propose the use of the fine fraction of settled sediments for assessment, which should primarily serve to screen out the non-hazardous sediments, and point at potentially hazardous ones, and not to be used as mandatory "pass/fail" environmental standards [43].

### 3.4. Fine Fraction vs. Bulk Sediment for Metal Contaminants

Comparing the results of metal concentrations in the fine and bulk fractions of the samples, a certain pattern was observed (Figure 6). Generally, all metals showed similar pattern—decreasing ratio of fine/bulk concentration with increasing share of the fine fraction (or decreasing share of the sand). For most of the river samples with sand content >60% concentrations in the fine fraction were at least 3 to 7 times higher than in the bulk, particularly for Cd, Cu, and Pb. Considerable difference was also noticed for sample K3, although its sand content was not so high (39.5). Such results were expected as it is well known that clay and silt tend to sequester high concentrations of metals due to commensurate increase in specific surface area [3,4,17].

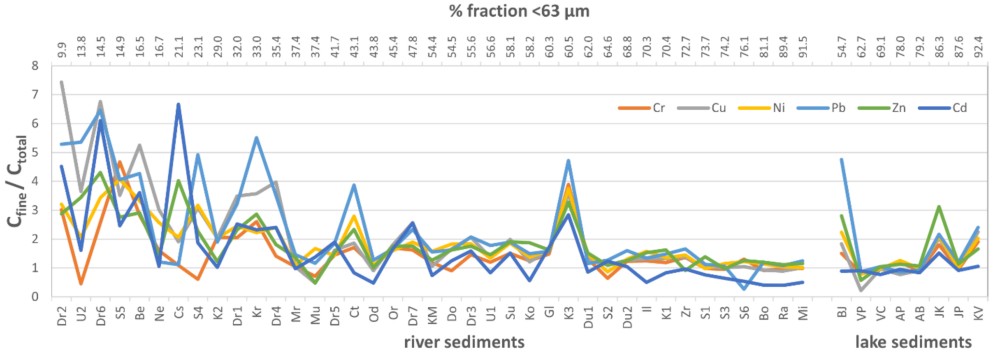

**Figure 6.** Metal concentration ratios between fine fraction and bulk sediment.

On the other hand, the ratio of fine/bulk concentrations for several samples with sand content >60% (K2, Mr, Mu), was not so high. This could be attributed to the fact that samples K2 and Mr had generally low concentrations of investigated metals, while sample Mu showed the opposite effect—higher concentrations of Cr, Cu, and Pb in the bulk than in the fine fraction. Such result is not uncommon since coarser particles show similar or even higher heavy metal concentrations in the vicinity of mining areas [11]; here it refers to the valley of the Drava River.

The difference between metal concentrations in the fine fraction and bulk of lake sediments was observed in samples BJ, JK, and KV. Although carbonates were the most abundant component in lake sediments, Al was more than twice as high in the fine fraction than in the bulk of these three samples (Figure 4A), which may indicate the presence of clays and explain the observed "anomaly".

There was also a reverse trend, i.e., higher concentrations of metals in bulk sediment than in the fine fraction. That goes for Cd in samples S6, S7, Il, Ko, Mi, Ra, and Od; Cr in samples S2, S4, U2, Mu; Ni in sample S2; Pb in sample S6; and Cu and Zn in sample Mu. The higher contamination of the bulk sediments was observed by Devesa-Rey et al. [2]; they suggested the formation of aggregates and metal-bearing coatings on sand particles at locations affected by contamination. Singh et al. [11] have also found that the presence of the nearby mining areas could be accounted for the higher concentration of the certain element in the coarse fraction. Similar explanation could be applied here for the samples pertaining to the river Sava (S2, S4, S6), Mura (Mu), and Ilova (Il) which could be considered as moderately polluted [39,41,44,45]. The abundance of Cr and Ni in the coarse fraction of the Sava River sediments and their low bioavailability [41] support the assumption that high concentrations of these metals are mainly of natural origin. Since the main contaminant load is usually connected to high water discharge [46], in the period of low water table fine-grained metal-bearing material is probably flocculated and nested between sand particles. When sieving was applied, these aggregates might have been removed along with the metals they contained. In addition, some elements are naturally present in source rocks (e.g., Cr, Ni, and Cd in samples S1, S2, Mi and Ra [47,48]) so they could have been found in higher concentrations in the bulk sediment.

As for the lake sediments, the only element found in higher concentration in the bulk sample was Cu in sample VP (Lake Vransko–Biog). This was probably related to agricultural activities around this area and high OM content (10.26%) in the bulk sample.

### 3.5. Processes Governing Metal Distributions

The principal component analysis (PCA) was carried out to explore the associations of investigated contaminants in the bulk and fine fraction of sediment samples (Figure 7). Following the clr-transformation, the first two principal components account for 82% of the total variability in bulk sediment samples (Figure 7A). PC1 is characterized by strong negative loading of clr(Ca) and positive loading of clr(Al) suggesting differentiation of sediments influenced by carbonates (Ct, BJ, JK, JP, KM, and KV) and those related to siliciclastic lithologies (Mu, Il, AP, AB, Or, and Kr). The positive PC2 axis yields strong loadings for clr(OM) and associated elements clr(Ni), clr(Cu), and clr(Cr) (Ra, Mi, S1, S2, and Gl). This implies that these elements are more associated with organic matter than the clay minerals [49]. The element assemblage of clr(Pb), clr(Zn), and clr(Cd) is mainly weighted by Drava (Dr1–Dr7) and Mura (Mu) rivers reflecting the mining and smelting activities in their catchments that date long back in history and could be held responsible for the contamination [39]. Interestingly, clr(Al) is positioned between these two element groups on the PC2 axis. Such position indicates that aluminosilicates are equally responsible for hosting both element associations. Similarly, the positioning of clr(Cd) can suggest its anthropogenic as well as geogenic origin, i.e., incorporation into the crystal lattice of carbonates [50].

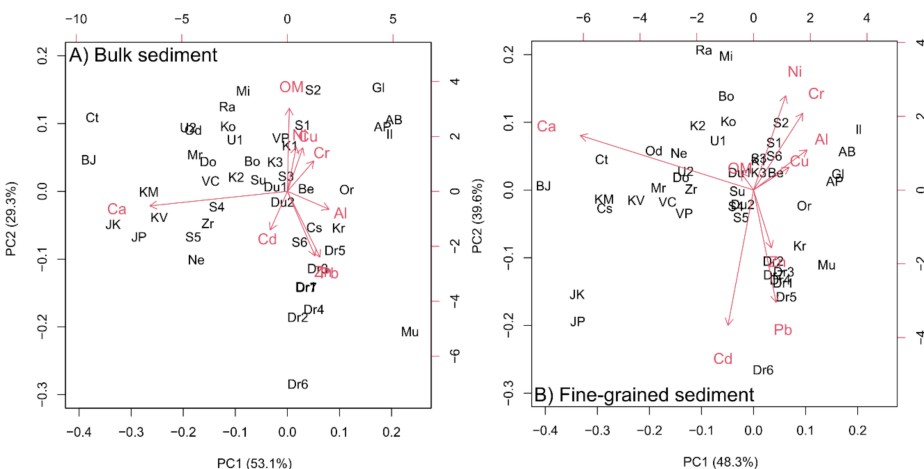

**Figure 7.** The principal component analysis (PCA) of sediments generated using the "provenance" package [51]: (**A**) bulk fraction, (**B**) fine fraction.

In the fine-grained sediments (Figure 7B), the element groups are more strictly separated at an angle of approximately 120°. The PC1 axis is dominated by strong negative loading of clr(Ca) which demonstrates again the influence of carbonate lithologies. The positive PC2 axis is characterized by gathering of clr(Al), clr(Cu), clr(Cr), and clr(Ni). Presumably, they associate with clay minerals and reflect the natural origin, which particularly pertains to Cr and Ni. Namely, this group is weighted with samples (Ra, Mi, S1, and S2) relating to catchments rich in the ultramafic component abundant in these elements [35,48]. The negative PC2 axis is influenced by the natural weathering of ore deposits and anthropogenic activities (Cd, Pb, and Zn) along the Drava, Mura, and Krapina rivers.

## 4. Conclusions

The distribution of six metal contaminants in the bulk (<2 mm) and fine fraction (<63 μm) of 47 freshwater sediments, sampled in 32 different sedimentary bodies, was compared to establish the basis for monitoring protocol. The common practice of geochemical normalization was not applied here since the lithological differences (particularly the abundance of carbonates) and spatial distribution of samples were too great to assure reasonable conclusion on the contaminants.

The samples containing >60% of sand fraction, particularly those from the river environments, had up to seven times higher concentrations of metals in the fine fraction than in the bulk sediment, but a significant difference was noticed for all samples containing more than 40% of sand. For most of those samples organic matter content was also found significantly higher in the fine fraction than in the bulk, indicating the organic matter preference for fine-grained sediments. Looking at these samples, the separation of the sand fraction and analysis of contaminants in the fine fraction in geochemical investigations seems advisable. However, some elements were found in higher concentration in the bulk than in the fine fraction, suggesting that a certain share of contaminants may be bound to OM-clay aggregates or naturally occurring in the sand fraction due to local lithological particularities or proximity of mining areas. When dealing with such environments, a careful inspection of both bulk and fine fraction is advisable.

In the case of lake sediments, the removal of sand fraction does not seem to be necessary, primarily because these are calm environments with fine-grained sedimentation. The accumulation of organic matter in such environments could also promote aggregation and virtual increase of the sand fraction and, consequently, the concentration of metals in it. The same could probably be applied for all calm environments with primarily fine sedimentation.

So, the answer to the question "should we separate the sand fraction for geochemical analysis?" is not straightforward yes or no. The preparation of samples for the geochemical

analysis should be carried out in accordance with the sedimentological characteristics of the investigated environment, which implies that these characteristics should be determined along with the geochemical analyses. This study indicates that it is generally advisable to analyze metals in the fine fraction of river sediments, while for the lake sediments either bulk or fine fraction can be used. It was also revealed that a prerequisite for a reliable assessment of the anthropogenic impact is knowing the geological setting of the investigated areas.

**Supplementary Materials:** The following are available online at https://www.mdpi.com/article/10.3390/min11060603/s1. Table S1: Location of sampling and particle size distribution parameters. Figure S1: Simplified geological map of Croatia.

**Author Contributions:** Writing—original draft preparation, N.V. Writing—data analysis, sampling, M.L. Supervision, N.M. Geochemical analysis, N.B. All authors have read and agreed to the published version of the manuscript.

**Funding:** This work was supported by "Croatian waters", legal entity for water management in Croatia, within the program "Monitoring of priority contaminants in biota and sediment in surface freshwaters of the Republic Croatia in 2019", contract number 10-050/19.

**Acknowledgments:** The authors are grateful to Kristina Pikelj for providing the geological map.

**Conflicts of Interest:** The authors declare no conflict of interest. The funders had no role in the design of the study; in the collection, analyses, or interpretation of data; in the writing of the manuscript, or in the decision to publish the results.

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
