# Peer review of "Partitioning of Metal Contaminants between Bulk and Fine-Grained Fraction in Freshwater Sediments: A Critical Appraisal"

_minerals, doi:10.3390/min11060603_

Round 1
Reviewer 1 Report
The article presents a critical assessment of the partitioning of metal contaminants between bulk and fine-grained fractions in freshwater sediments.
This study is aimed at:(1) investigation of the distribution of selected heavy metals (Cd, Cr, Cu, Ni, Pb and Zn) in the bulk (<2 mm) and fine-grained fraction (<63 μm) of freshwater sediments from a number of different environments; (2) evaluation of differences in metal concentration between those two fractions and their relation to sediment characteristics (particle size distribution and organic matter content) and (3) bringing the potential guidance for future environmental and monitoring studies about the choice of sediment fraction to be analysed while assessing metal contamination in freshwater bodies.
The manuscript is relevant to the scope of the Minerals, is scientifically sound and valid. When preparing the article, all the rules for journals were followed. The structure of the reviewed article is well-thought-out and clear. The content of the deliberations described in the paper is appropriate for the subject of research and the specific objective. Description, analysis and interpretation are presented in a legible way. The conclusions drawn by the authors are supported by the data provided in the text. The quoted literature is a useful supplement for the recipient interested in further exploring the issues.
However, minor corrections should be introduced: - in the text descriptions of Figs. 2 and 3 there are a and b, and in the figure capital A and B are used; - why the figure 4 has been split into parts a and b? - analysing the data presented in Figure 2 concerning the correlation analysis, it would be necessary to describe what test was used for statistical hypotheses verification? Were the test assumptions such as the distribution of variables met by the results used in the calculations? At what significance level the coefficients of linear correlation r were tested?
Author Response
The authors are grateful for the thorough revision of the manuscript. The comments were carefully considered and answered in detail below, and all the suggestions were taken into account. All the changes in the manuscript were made accordingly and highlighted in the text.
Reviewer #1
The article presents a critical assessment of the partitioning of metal contaminants between bulk and fine-grained fractions in freshwater sediments.
This study is aimed at:(1) investigation of the distribution of selected heavy metals (Cd, Cr, Cu, Ni, Pb and Zn) in the bulk (<2 mm) and fine-grained fraction (<63 μm) of freshwater sediments from a number of different environments; (2) evaluation of differences in metal concentration between those two fractions and their relation to sediment characteristics (particle size distribution and organic matter content) and (3) bringing the potential guidance for future environmental and monitoring studies about the choice of sediment fraction to be analysed while assessing metal contamination in freshwater bodies.
The manuscript is relevant to the scope of the Minerals, is scientifically sound and valid. When preparing the article, all the rules for journals were followed. The structure of the reviewed article is well-thought-out and clear. The content of the deliberations described in the paper is appropriate for the subject of research and the specific objective. Description, analysis and interpretation are presented in a legible way. The conclusions drawn by the authors are supported by the data provided in the text. The quoted literature is a useful supplement for the recipient interested in further exploring the issues.
However, minor corrections should be introduced: - in the text descriptions of Figs. 2 and 3 there are a and b, and in the figure capital A and B are used; - why the figure 4 has been split into parts a and b? - analysing the data presented in Figure 2 concerning the correlation analysis, it would be necessary to describe what test was used for statistical hypotheses verification? Were the test assumptions such as the distribution of variables met by the results used in the calculations? At what significance level the coefficients of linear correlation r were tested?
Corrections:
- The capital letters were inserted in figure caption and in the text.
- The figure 4 was split in two because it was too large for one page – it seemed more convenient. We have changed that.
- The numbers of figures were also corrected since there was a mistake in the previous version of the manuscript; some figures were also slightly changed.
- We tested our data for normality using Shapiro-Wilk’s test. Aluminum only showed the p-value of 0.58 implying that its distribution is not significantly different from the normal distribution. For others, the distribution is heavily skewed. Even though the content of clay and grain size did not follow the normal distribution, the mutual relationship between Al and clay content was examined using linear regression analysis. In our case it gives more sense. The level of significance was 0.001 and obtained p-values for regression analysis were 6 x 10-6 suggesting that our model was reliable.
Reviewer 2 Report
Some ideas relevant for this work have been discussed in Grygar and Popelka (2016) as well as in critical comments to works by Pavlovič et al. (TM Grygar 2016, Sci Tot Environ; and 2020, Catena). In fact, I wrote those papers to avoid repeating facts in peer reviews and thus I quoted them in now in this peer review. IT IS NOT INSTRUCTION TO CITE THEM, better if authors checked whether anything of their content could of interest of them, and pay attention to the references quoted there. The references substantiate why much more careful attention must be paid to geological map and sampling strategy. Geological info should be added to figures and supplementary table.
Geochemical normalisation is not assuming everything is in the mud fraction (clay+silt), but finding in what the risk elements are bound – it can be organic matter, Fe- and/or Mn-oxides, but also ore mineral grains. Definitely, grain size is not the only thing to be considered. Some aspects related to lithological correction of sediment composition are dealt with in the references 1st paragraph of this review.
Monitoring of contamination was nicely dealt with by Mokwe-Ozonzeadi et al. (2019), the authors of the manuscript under review should really read it. SPM sampling is indeed the best option for the purpose addressed in the manuscript, the samplers can be really simple, or adequate fine sediment can be always found in the river channels, to prevent horrible phrases like “<63 μm-sized particles, is in rather short supply” (lines 103-104) – if one is not afraid of going along the bank or a bit farther from bank to channel, there are always traps of fine sediments, in between stones, embankments, debris, etc. We used that approaches several times. Still, SPM contains a bit more contamination, Hošek et al. (2020) in Environ Geochem Health.
Sieving – this topic is dealt with in Tůmová et al. (2019) in J. Soils Sediments. Some aspects related to sieving are dealt with in the references in the 1st paragraph of this review.
Regarding CoDA using logratio methodology, clr is not necessarily a good idea, as mentioned by M. Greenacre (2019) in Mathematical Geosci – results of clr analysis are not necessarily interpretable in real world sense. Most papers declaring to apply Aitchinson statistics to real geochemistry problems suffers of gap between mathematicians and geoscientists, first the purpose of the statistical analysis must be specified and then the logratio approach selected. Typical problem of clr with all elements (and not with specifically selected sub-composition) is destruction of useful geochemical signals, in which, e.g., geochemical normalisation is included by having macrocomponents including those diluting risk elements, but it is done by unclear manner of geometric means in clr formula. Relarding PCA, I would also point attention of authors to Reid and Spencer (2009) in Env Poll. The authors could show PCA with clr but also with raw concentrations with outliers removed (or using robust PCR), otherwise the PCR interpretation is risky. Some aspects related to miraculous black box of PCA are dealt with in the references in the 1st paragraph of this review.
The authors should consider sand is not always quartz!! The sand composition (including its Al content) its provenance specific. Provenance must be highlighted in graphs. Plots of element concentrations against size fractions shuld be sarated tio mafic and felsic catchments, at least, see e.g. Laceby et al. (2015) in J Soils Sediments. Some aspects related to sediment provenance are dealt with in the references in the 1st paragraph of this review.
I would recommend authors to use Lake or lake instead of jezero or j, better for non-Slavonic readers.

Author Response
The authors are grateful for the thorough revision of the manuscript. The comments were carefully considered and answered in detail below, and all the suggestions were taken into account. All the changes in the manuscript were made accordingly and highlighted in the text.
All the answers to reviewer can be found in the attached file

Round 2
Reviewer 2 Report
I thank authors for their effort and for considering my comments. In some cases the responses were a bit like misunderstanding.
The geological map added to the supplement does not show ultramafic provenance - I do not see it in the legend, for example. Geogenic anomalies (provenances) are still not shown in figures with risk element concentrations. The relationship between element concentrations and geology cannot thus be traced from the data presentation. OK, depends on the authors if they wish that, but I would prefer the authors improve this.
I do not understand the arguments against geochemical normalisation. If CaCO3 is one of the main drivers of sediment composition variability, its influence should be eliminated before judging impact of grain size (sieving or bulk) or possible contamination, either natural or anthropogenic. The authors state also fine fraction can be dominated by carbonates. OK, for this either re-calculation to carbonate-free basis can be done, or geochemical normalisation to a component (components) definitely not proportional to carbonate content (and preferably geochemically interrelated with risk elements) could be advantageous. Both, of course, only if carbonates do not contain risk elements. The authors did not attempt to check it. The lack of correlation of fine fraction with Al is not reason to omit normalization – and as some of the papers I mentioned in my preceding review show, normalisation is not only to perform grain size correction, but also to eliminate matrix dilution effects – here dilution by carbonates. To find whether the normalisation is meaningless, the authors could have compared fine/bulk ratio for raw concentrations, fine/bulk ratio for carbonate-free basis, and/or fine/bulk for geochemically normalised concentrations. Of course, depends on the authors.
But, in revised version the authors quoted Reimann et al. and their scepticism regarding EF. Reimann refused that concept using pretty unfair examples – like EF cannot work in comparison of litter and mineral soil. Sure, it was Reimann’s choice, as well as the choice of authors of the manuscript under review to follow this peculiar example, of course, everybody has a freedom of choice. But what matters here – how do authors consider variable carbonate content in their data evaluation? I can, for example, say: what about if the fine fractions of their samples contain less carbonates and thus also more risk elements - is it reason for preferring fine fraction? Still, the repetition of Reimman’s approach has not much sense here. The authors should reconsider whether recalling Reimann bias is necessary here.
The same matter is the discussion on PCA. I guess the authors did not get my message. The aim of PCA should be separation of major geochemical signals, best if the signals are qualitatively known in advance and the geochemical data are treated in a manner appropriate to that particular qualitative knowledge. The percentage of variability explained by the first two components has no relevance in this case, and any shape of element loading cannot replace the fundamental qualitative judgement before before choice of alr or clr or so for PCA. Let me summarise what is in the manuscript and authors’ reply: In the presented dataset, at least three factors are relevant – carbonate content, ultramafic provenance, and contamination. They declared these factors explicitly. Perhaps also quartz-bearing rocks are at play – with silt-sized quartz, also mentioned in the manuscript – loess and similar Quaternary deposits. What is purpose of showing two components for clr? Which of those above-mentioned components are visualised? What is effect of clr on expression of those components knowing Ca is a part of the denominator (within geometrical mean of all concentrations) although it is probably indirectly proportional to all other elements and will thus amplify the carbonate-dilution effect instead of filtering it and visualise provenance or contamination? How the authors know that the proximity of Ni/Al and Zn/Al loadings is non-sense? (they used this argument to judge suitability of clr in their reply to my peer review). The authors showed in their reply PCA without outliers – were the outliers contaminated sediments? Carbonate rich ones? Those with ultramafic provenance? I still do not see for what purpose the PCA is presented. For example, all plots indicate association of Ca and Cd – is Cd in carbonates? This kind of approach is somewhat wasted occasion to really utilise PCA and contribute to discussion of logratio methodology. The arguments provided by the authors are slightly ridiculous. I would recommend the authors tu really use PCA and get som real interpretation from the plots, or remove it as irrelevant.
In sum: The dataset promoted by the authors deserves to be published. I tried to provoke the authors to think more about geochemical data mining, the result of my effort is addition of references instead of some action. I apologise for the horribly huge number of spelling errors in my preceding report, which perhaps devaluated my intentions. I still think publication of any new geochemical dataset can be an occasion to corroborate and test (discuss) the data processing tools, implement provenance and lithology corrections. Nothing personal. I am sorry for this.
Author Response
Please find the answers attached.
